# Nutrition Management in Older Adults with Diabetes: A Review on the Importance of Shifting Prevention Strategies from Metabolic Syndrome to Frailty

**DOI:** 10.3390/nu12113367

**Published:** 2020-11-01

**Authors:** Yoshiaki Tamura, Takuya Omura, Kenji Toyoshima, Atsushi Araki

**Affiliations:** Department of Diabetes, Metabolism, and Endocrinology, Tokyo Metropolitan Geriatric Hospital, 35-2 Sakae-cho, Itabashi-ku, Tokyo 173-0015, Japan; tamurayo@tmghig.jp (Y.T.); takuya_omura@tmghig.jp (T.O.); kenji_toyoshima@tmghig.jp (K.T.)

**Keywords:** diabetes mellitus, nutrition management, elderly, frailty, cognitive impairment

## Abstract

The increasing prevalence of older adults with diabetes has become a major social burden. Diabetes, frailty, and cognitive dysfunction are closely related to the mechanisms of aging. Insulin resistance, arteriosclerosis, chronic inflammation, oxidative stress, and mitochondrial dysfunction may be common mechanisms shared by frailty and cognitive impairment. Hyperglycemia, hypoglycemia, obesity, vascular factors, physical inactivity, and malnutrition are important risk factors for cognitive impairment and frailty in older adults with diabetes. The impact of nutrients on health outcomes varies with age; thus, shifting diet therapy strategies from the treatment of obesity/metabolic syndrome to frailty prevention may be necessary in patients with diabetes who are over 75 years of age, have frailty or sarcopenia, and experience malnutrition. For the prevention of frailty, optimal energy intake, sufficient protein and vitamin intake, and healthy dietary patterns should be recommended. The treatment of diabetes after middle age should include the awareness of proper glycemic control aimed at extending healthy life expectancy with proper nutrition, exercise, and social connectivity. Nutritional therapy in combination with exercise, optimal glycemic and metabolic control, and social participation/support for frailty prevention can extend healthy life expectancy and maintain quality of life in older adults with diabetes mellitus.

## 1. Introduction

The increase in diabetes prevalence in the elderly has led to a greater understanding of geriatric diabetes care and the fundamentals of management. Populations are aging across all countries within the Organization for Economic Cooperation and Development as well as in China, India, Brazil, and Russia. Although the pace of population aging differs among countries, the ratio of adults over 65 years of age to those at working age is expected to almost double in the next 40 years across most of the world [1]. Japan is a super-aging society, with one-third of its citizens already over the age of 65 years. As a result, a substantial increase in diabetes prevalence over the next few decades is expected in Japan [2]. Similarly, an increase in the number of older patients with diabetes is observed in the United States [3]. This increase in the rate of older individuals with diabetes requires the management of those with physical or cognitive impairment who fail to accomplish diabetes-related self-care such as insulin injection. Thus, aging of the global population is associated with the emergence of diabetes management in the elderly as an important public health issue of the 21st century.

Although novel therapeutic agents for diabetes have broadened our treatment options, a balanced diet and regular exercise are essential to patients’ health. Recently, nutrition research has made progress, and the era in which aging itself becomes a therapeutic target has begun; therefore, this review aims to provide an update on the importance of proper nutrition in older people with diabetes. Aims of modern diet therapy are not only properly managing glycemic control but also preventing frailty, sarcopenia, and dementia, to provide optimal medical care for each patient. Based on the latest evidence, we would like to propose “shifting prevention strategies from metabolic syndrome to frailty in older adults with diabetes”.

### 1.1. Frailty and Cognitive Impairment in Older Adults with Diabetes Mellitus

The assessment and maintenance of functional ability are extremely important in the care of older adults with diabetes. The American Diabetes Association [4], the Japan Diabetes Society/Japan Geriatrics Society [5], and the International Diabetes Federation Working Group [6] recommend setting glycemic control targets and emphasize that consideration should be given to physical and cognitive functions.

### 1.2. Frailty

Diabetes and frailty are intricately linked; furthermore, diabetes is strongly associated with reduced mobility and activities of daily life (ADL) [7]. Frailty is also associated with increased risks of falls, disability, institutional admission, and mortality due to age-related decreases in reserve capacity of organ function. Interventions by diet and exercise and optimal treatment of diseases can partially reverse frailty to robustness. The Cardiovascular Health Study (CHS) criteria proposed by Fried et al. constitute a well-established diagnostic index for frailty that focuses on parameters of physical function [8,9]. In the CHS, the prevalence rates of diabetes were 18.8% in individuals without frailty, 24.5% in those with prefrailty, and 32.4% in those with frailty [9]. An observational study of Beijing residents over 55 years of age revealed that diabetes increased the prevalence and incidence of frailty 1.36- and 1.56-fold, respectively [10]. Several studies show that the mortality is higher in patients with diabetes and frailty and that frailty is an independent risk factor for mortality [11,12]. Hyperglycemia, hypoglycemia, low hemoglobin A1c (HbA1c), insulin resistance, abdominal obesity, cardiovascular disease, low physical activity, and malnutrition are associated with frailty in individuals with diabetes.

### 1.3. Cognitive Impairment

Diabetes mellitus is intricately linked to cognitive impairment and dementia. In a meta-analysis of 144 prospective studies, diabetes conferred an approximately 1.5- to 2.0-fold increased risk for cognitive impairment and dementia [13]. Cognitive impairment in patients with diabetes mellitus tends to affect not only verbal and visual memory but also attention, information processing ability, and executive function [14]. The risks for vascular dementia and Alzheimer’s disease in diabetes are increased by approximately 100% and 50%, respectively [13]. A combined analysis of 14 studies examining data from 2.3 million people across Asia, the Americas, and Europe revealed that patients with diabetes had a 60% greater risk for dementia [15]. Midlife development of diabetes is associated with a 19% increase in cognitive decline over 20 years [16], and obesity in midlife is also a significant risk factor for dementia [17]. Hyperglycemia underpins the association between metabolic syndrome (MetS) and cognitive impairment [18]. Importantly, impairment of memory and executive function leads to poor adherence to diabetes-related self-care activity. In particular, impairment of executive function adversely affects instrumental ADL (IADL; shopping, meal preparation, and money management) and leads to poor adherence to diet and medication.

However, not all patients with diabetes are prone to dementia, and untreated diabetes is an important risk factor for dementia. In a longitudinal study of 1289 patients with type 2 diabetes who participated in Alzheimer’s Disease Neuroimaging Initiative, the untreated diabetes group, defined as those with a fasting blood glucose of ≥ 126 mg/dL, had a 1.6 times higher risk of dementia than the normoglycemia group, defined as those with a fasting blood glucose of < 100 mg/dL, indicating that the risk of dementia was not significantly higher in patients treated with one or more hypoglycemic agents [19].

Studies have led to the identification of several risk factors for dementia and cognitive decline in adults with diabetes that can be classified into several groups: (a) factors associated with blood glucose control: hyperglycemia, severe hypoglycemia [20,21], and fluctuations in blood glucose level [22,23]; (b) factors associated with cardiovascular disease: cerebrovascular disease, peripheral artery disease, chronic kidney disease, albuminuria, systolic blood pressure, and high chronic kidney disease levels [24,25]; and (c) lifestyle factors: reduced leisurely activities and poor social network [26], low body mass index (BMI), weight loss, weight gain [27], low intake of carotene, vitamin B2 and green vegetables [28]. These reports suggest that treating diabetes with attention to risk factors may lead to the prevention of dementia.

### 1.4. Interaction between Frailty and Cognitive Impairment

Several observational studies have confirmed the association between frailty and cognitive decline or dementia. A five-year prospective study of 2305 individuals aged 70 years and older showed that frailty based on the CHS criteria, FI-CGA (Frailty Index-Comprehensive Geriatric Assessment, and clinical frailty scale were all risk factors for cognitive decline [29]. In a meta-analysis of seven studies, physical frailty based on the CHS criteria was associated with a 1.28-fold increased risk of Alzheimer’s disease and a 2.70-fold increased risk of vascular dementia than in robust individuals among older adults [30].

On the other hand, cognitive decline is also a risk factor for frailty. In a Korean cross-sectional study of 10,388 individuals with cognitive impairment assessed by the mini-mental state examination, male and female patients with cognitive impairment were 1.81 and 1.69 times more likely to have frailty, respectively [31]. At the one-year follow-up, patients with mild or moderate Alzheimer’s disease were more likely to develop frailty. In particular, decreased executive function, as assessed by the trail-making and Stroop tests, was shown to be associated with decreased walking speed and prolonged time on the timed “up and go” test [32].

A decline in physical function may lead to inactivity or isolation, therefore negatively impacting cognitive function. Conversely, a decline in cognitive function, including executive function, adversely affects IADL and physical function. Together, physical and cognitive impairment is a strong prognostic factor for mortality in patients with diabetes [33]; therefore, frailty and cognitive function should be considered in diabetes care for older adults. Based on the above, the concept of cognitive frailty, which combines physical frailty with cognitive impairment that has not yet led to dementia, has been proposed [34]. Albeit multifactorial, frailty and cognitive impairment share a common etiology; therefore, cognitive frailty is an important concept that should be considered in the prevention of dementia.

## 2. Mechanistic Insights into Diabetes, Frailty, and Cognitive Impairment

The etiology, pathophysiology, and metabolic control of diabetes, regardless of whether it is type 1 or type 2, are considered to share common traits with frailty and cognitive impairment. Reduced insulin secretion, insulin resistance, arteriosclerosis, chronic inflammation, oxidative stress, mitochondrial dysfunction, poor glycemic control, decreased physical activity, and malnutrition in diabetes are potential causes of frailty and cognitive impairment observed in these patients (Figure 1).

### 2.1. Insulin Resistance

Insulin resistance, i.e., reduction in cellular responsiveness to insulin, is closely associated with other features of MetS, including diabetes and dyslipidemia [35]. In healthy individuals, skeletal muscle accounts for 40–50% of lean body mass and is the major target tissue for insulin. Insulin sensitivity is required for normal glucose disposal and regulation of lipid oxidation. In vitro and in vivo models have demonstrated that insulin resistance contributes to the pathogenesis of sarcopenia. Impaired insulin signaling can result in reduced protein synthesis, thereby altering anabolic response to exercise, amino acids, and insulin [36,37].

In older patients with MetS, insulin resistance negatively affects cognitive function [38,39]. Studies investigating the mechanisms underpinning the association between Alzheimer’s disease and insulin resistance in animal models and humans reveal that disrupted insulin signaling due to the phosphorylation of insulin receptor substrate-1 (IRS-1) at serine residues causes the formation of amyloid plaques and neurofibrillary tangles [40].

### 2.2. Arteriosclerosis and Brain White Matter Lesions

Diabetes mellitus and cerebrovascular disease are closely related, and the strong association between arteriosclerosis and cognitive function has been extensively demonstrated. Oxidative stress, advanced glycation end-products, polyol, and protein kinase C activation contribute to vascular endothelial dysfunction, which may result in atherosclerosis and diabetic macroangiopathy [41].

The link between arteriosclerosis and frailty may be important based on the increased risk of cardiovascular events that negatively impact ADL in patients with arteriosclerosis. Furthermore, arteriosclerosis can directly affect frailty. Multiple studies have shown significant associations of vascular dysfunction with low muscle mass or strength [42]. Peripheral chronic ischemia with impaired oxygen supply due to arteriosclerosis can induce muscle atrophy [43]. However, the effects of arteriosclerosis on myocytes are not well known. In particular, further investigation is necessary to elucidate the role of myokines, NADPH oxidases, and mitochondrial changes in arteriosclerosis.

Cerebral white matter lesions can occur as a result of chronic ischemic changes in the brain. Diabetes is a well-known risk factor for cerebral infarction and is associated with increased risk of vascular dementia, with recent studies highlighting the adverse effects of diabetes on cognitive function via cerebral white matter lesions [44,45]. White matter hyperintensities are well detected in patients with diabetes [46]. The accumulation of damage in cerebral microvessels can lead to the progression of white matter hyperintensities by mechanisms shared with arteriosclerosis. A link between white matter hyperintensities and impairment in cognitive function or IADL has been shown in older adults with diabetes [45]. A recent study using brain magnetic resonance diffusion tensor imaging reported that alterations in white matter integrity in the anterior thalamic radiation were associated with sarcopenia in patients with diabetes [47].

### 2.3. Chronic Inflammation, Oxidative Stress, and Mitochondrial Dysfunction

Chronic inflammation is strongly associated with age-related pathologies including cardiovascular diseases, diabetes, frailty, and dementia. Inflammatory cytokines are elevated in obesity and overnutrition [48], and aging is associated with chronic low-grade inflammation [49]. Loss of β-cell function and deficiency in insulin secretion drive hyperglycemia and metabolic dysfunction, which impose a metabolic burden on the liver, skeletal muscle, and adipose tissue. These changes in insulin action can be induced by inflammatory cytokines, including tumor necrosis factor alpha and interleukin-6 (IL-6), whereas hyperglycemia can promote further inflammation [50]. The ensuing chronic inflammation can induce anabolic resistance in muscle, resulting in sarcopenia due to reduced insulin-like growth factor 1 (IGF-1) production and signaling that are essential for muscle regeneration and maintenance [51]. The link between immunity and metabolism is well established; however, inflammation-driven neurodegeneration has also been garnering increasing attention. In experimental models, chronic inflammation has been demonstrated to adversely affect the brain and nerves. Perpetuation of microglial activation due to persistent exposure to proinflammatory cytokines causes functional and structural changes, which eventually result in pathological accumulation of amyloid beta, although inflammatory responses can be both beneficial and detrimental to the brain [52].

Oxidative stress arises due to an imbalance between the production and processing of reactive oxygen generated by endogenous or extrinsic factors; oxidative stress also plays important roles in the pathogenesis of lifestyle-related diseases including diabetes. Numerous lines of evidence from in vivo and in vitro studies have demonstrated that oxidative stress is elevated in the hyperglycemic state and in patients with diabetes and can lead to various diabetic complications [53]. Within the human body, neurons are exquisitely susceptible to damage by free radicals [54], and individuals with cognitive dysfunction have increased oxidative stress in the central nervous system [55]. Similarly, oxidative stress can induce skeletal muscle atrophy and sarcopenia via mitochondrial dysfunction [56]. Reactive oxygen species, which are dominantly produced by mitochondria [57], are in a physiological balance to maintain muscle volume and function. As mitochondrial DNA is susceptible to damage by oxidative stress [58], excessive oxidative stress in the context of aging and diabetes can have adverse effects in the muscle [59,60]. In addition, mitochondria and neuroplasticity are intricately linked and mitochondrial dysfunction is observed in early Alzheimer’s disease pathogenesis [61]. Thus, mitochondrial dysfunction and oxidative stress might be important pathogenic mechanisms underlying dementia, sarcopenia, and frailty in diabetes.

### 2.4. Hyperglycemia and Hypoglycemia

Hyperglycemia was associated with cognitive decline. The negative impact on cognitive performance was highly individualized, and approximately 50% of individuals with diabetes demonstrated such hyperglycemia-related cognitive disruption [62]. Activation of the polyol pathway, increased formation of advanced glycation end-products, diacylglycerol activation of protein kinase C, and increased glucose shunting to the hexosamine pathway, all of which are altered in response to hyperglycemia, can impair neuronal function [63]. Conversely, older adults with diabetes are at greater risk of hypoglycemia due to multiple comorbidities such as malnutrition, polypharmacy, and renal dysfunction [64,65]. Similar to hyperglycemia, hypoglycemia can also cause neuronal death and cognitive impairment via excitotoxicity and DNA damage [66]. Severe hypoglycemia can lead to neuronal cell death in the central nervous system due to damaged mitochondria, elevated intracellular Ca^2+^ levels, and increased release of inflammatory cytokines. Repeated periods of mild hypoglycemia can cause axonal degeneration in the peripheral nervous system due to delayed motor conduction and afferent neurodegeneration from the peripheral to central nervous system in a dying-back pattern [67].

Increased fluctuations in blood glucose level may be associated with cognitive impairment and decline. In a 21-year longitudinal study, cognitive decline was highest in patients with diabetes and an HbA1c of ≥ 7.0% among those with large fluctuations in blood glucose levels [23]. Quantification of brain white matter lesions in our older patients with diabetes by the SNIPER software showed an increase in brain white matter lesions in those with cognitive impairment (≤26 points in the mini-mental state examination) or IADL decline [45]. In multivariate analysis, the glycoalbumin/HbA1c ratio, an index of blood glucose fluctuations, was independently associated with brain white matter lesions, suggesting that large fluctuations in blood glucose levels might lead to white matter lesions, cognitive dysfunction, and IADL decline.

Recent studies have highlighted the relationship between blood glucose control and frailty. In addition, poor glucose control is associated with sarcopenia as well as cognitive decline [68]. Hyperglycemia induces muscle atrophy and protein catabolism by blocking ubiquitin-dependent degradation of the transcription factor KLF15 involved in skeletal muscle loss [69]. Although the mechanism is not yet known, hypoglycemia is expected to negatively affect muscle homeostasis as fluctuations in blood glucose induce oxidative stress [68].

### 2.5. Physical Inactivity

Physical inactivity is an independent risk factor for dementia and frailty. Lack of exercise leads to the accumulation of visceral fat; as a source of chronic inflammation, increased visceral fat may be involved in the pathogenesis of insulin resistance, atherosclerosis, cognitive impairment, and frailty. The protective effect of exercise may be ascribed to the improvement in insulin resistance and inflammation mediated by myokines or the reduction in visceral fat mass by regular exercise. Muscle contraction produces the myokine IL-6, which activates AMP-activated protein kinase (AMPK) and phosphoinositide 3-kinases and enhances glucose uptake and fat oxidation in skeletal muscle [70].

Regular exercise is beneficial for reducing cardiovascular risk factors involved in the pathogenesis of dementia. In addition, exercise may improve insulin resistance, inflammation, and oxidative stress associated with impaired cognition. Furthermore, exercise produces myokines (cathepsin-B, irisin), which stimulate the production of brain-derived neurotrophic factor [71].

## 3. Nutrition and Nutrients in Older Adults with Diabetes Mellitus

### 3.1. Obesity: Setting an Appropriate BMI Range

Obesity in older patients with diabetes has been increasing globally. In a prospective cohort study of Japanese individuals, an ethnic group well known for the low prevalence of obesity, the prevalence of obesity, defined by a BMI of ≥ 30 kg/m^2^, was increased by 1.5 times over 12 years from 3.5% in 2000 to 5.4% in 2012 [72]. Caution is warranted in evaluating BMI, defined as body weight (kg) divided by height squared (m)^2^, since body height may be shortened in older subjects, leading to the overestimation of BMI in these individuals.

Obesity is a well-known risk factor for cardiovascular diseases [73]; however, whether this is also true for older patients, especially for those with diabetes, is yet to be fully clarified. Most of the studies investigating older individuals with and without diabetes have failed to demonstrate a positive correlation between obesity and cardiovascular events or mortality. For example, a study in China showed that BMI was associated with CHD incidence in younger females (40–≤55 years old), but not in older subjects (>55 years old) [74]. In another observational study in Spain on subjects at high cardiovascular risks (aged 55–80 years), BMI was not associated with all-cause mortality, whereas waist-to-height ratio and waist circumference were associated with all-cause mortality [75]. Although studies of older patients with diabetes are scarce, the results of the Japanese Elderly Intervention Trial (J-EDIT) provide several meaningful findings. The J-EDIT was originally planned to validate the effect of multiple interventions aimed at diabetes, hypertension, dyslipidemia, and obesity on diabetic microvascular complications, macrovascular events, cognitive impairment, and physical disability. Although the trial failed as an interventional study since the HbA1c levels did not show a meaningful difference, it provided valuable information as an observational study revealing potential risk factors for the outcomes indicated above [76]. In their assessment of the relationship between BMI and six-year mortality in the participants of the J-EDIT, Tanaka et al. found that those with a BMI of < 18.5 kg/m^2^ were at a higher risk of mortality. The lowest mortality risk was observed in those with a BMI of 18.5–24.9 kg/m^2^, but the risk was not increased in those with a BMI of ≥ 25 kg/m^2^ [77]. Furthermore, studies reported that among patients who experienced cardiovascular events, the risk for the next event or death was lower in obese patients compared to those with normal weight, in a phenomenon termed the “obesity paradox” [78].

Regarding the effect of obesity on decline in ADL, several meta-analyses and systematic reviews have reported that a high BMI has a negative impact on physical function. Schaap et al. showed that those with a BMI of ≥ 30 kg/m^2^ were at a higher risk of functional decline including gait disturbance and difficulty in stair climbing [79]. Conversely, low BMI is also a risk for frailty. In a Japanese cross-sectional study investigating the association between BMI and prevalence of frailty, the authors showed a U-shaped relationship, with the lowest risk of frailty observed in individuals with a BMI of 21.4–25.7 kg/m^2^ [80]. Few studies have investigated the association between obesity and ADL decline in older patients with diabetes. García-Esquinas et al. reported that the positive risk for frailty in older patients ≥ 60 years of age was attenuated by adding obesity as a covariate; the authors concluded that abdominal obesity had a partial influence on the progression of frailty in these individuals [81]. Intensive lifestyle intervention and weight reduction in the Look AHEAD (Action for Health in Diabetes) study delayed disability onset and extended the disability-free life expectancy in older individuals with diabetes [82].

High BMI has a protective effect in cognitive impairment in older patients. This is also true for patients with diabetes. In a Korean cohort of patients aged ≥ 40 years, a lower BMI was associated with a higher incidence of dementia [27].

These findings altogether suggest that high BMI (≥30 kg/m^2^) should be corrected to prevent the progression of functional disability and that low BMI (<18.5 kg/m^2^), which is also a diagnostic criterion for malnutrition [83], should also be avoided in older adults with diabetes.

### 3.2. Metabolic Syndrome

MetS is characterized by visceral fat accumulation accompanied by multiple metabolic disorders including hypertension, glucose intolerance, and dyslipidemia. Importantly, visceral fat is increased whereas lean body mass is decreased in older subjects and that those with large waist circumference or with MetS, accompanied with normal BMI are increased in older people.

To date, various diagnostic criteria have been proposed for MetS. Caution is warranted since some guidelines, including those of the International Diabetes Foundation, require large waist circumference as a minimum requirement whereas others, such as those of the American Heart Association and the National Heart, Lung, and Blood Institute (AHA/NHLBI), do not [84]. In developed countries, the prevalence of MetS increases with age. A recent report from the Ministry of Health, Labor, and Welfare of Japan demonstrated that the rate of individuals with highly suspicious MetS was highest in those aged 70 years or over for both sexes [85]; in that report, glucose intolerance was defined as an HbA1c ≥ 6.0% and triglyceride level was excluded from the definition of dyslipidemia.

MetS is a strong risk factor for progression to atherosclerosis and cardiovascular events in individuals with and without diabetes [86]. However, few studies have investigated this association in older patients with diabetes. Hiller et al. reported that MetS increased the risk for total and cardiovascular mortality in older women with diabetes [87]. However, Monami et al. reported that stratified analysis by age did not reveal the influence of MetS on cardiovascular mortality in patients older than 70 years of age with type 2 diabetes [88]. In the J-EDIT, Sakurai et al. reported that MetS, which was diagnosed based on the AHA/NHIBI criteria, was associated with increased prevalence of coronary heart disease and stroke [89]. However, the study was cross-sectional and most of the patients were under the age of 75 years.

The impact of MetS on ADL decline is controversial, and few studies have investigated the association between MetS and ADL decline in older patients with diabetes. In a prospective cohort study of Mexican Americans, with an average cohort age of 70.6 years, the participants were classified by their diabetes status; the authors reported that a significant progression of ADL disability was observed in participants with diabetes in association with MetS (defined by the National Cholesterol Education Program Adult Treatment Panel [NCEP ATP] III criteria) [90]. However, the impact of diabetes itself on ADL decline was far stronger and the additive effect of MetS on diabetes was minimal. In the J-EDIT, Sakurai et al. reported that MetS diagnosed by the AHA/NHIBI criteria was a significant risk for basic ADL decline in older patients with diabetes [91]. On the other hand, the InCHIANTI study, a large prospective Italian cohort study, reported that in patients aged 74 years or over with MetS (according to the NCEP ATP III criteria), the risk of incident frailty was reduced [92].

Several observational studies have indicated that, unlike simple obesity, MetS might be a risk factor for cognitive impairment in older patients [93,94]. The risk for atherosclerosis and chronic inflammation associated with MetS might have influenced these results. However, most of the studies included individuals younger than 75 years of age, and evidence for the impact of MetS on cognition in those over 75 years of age is limited. Besides, evidence is scarce on patients with diabetes and MetS [95]. Future studies should target older patients with diabetes and MetS.

In summary, prevention of MetS is recommended in patients with diabetes younger than 75 years of age, although evidence supporting MetS prevention in older old patients (age ≥ 75 years) with diabetes is lacking.

### 3.3. Sarcopenic Obesity

Sarcopenic obesity is defined by the concomitant presence of sarcopenia and obesity [96]. The various diagnostic criteria that have been proposed and used for sarcopenic obesity hinder the effective comparison and meta-analysis of various studies conducted on sarcopenic obesity [96]. Moreover, some studies might have utilized only muscle mass loss or muscle weakness (dynapenia) as the definition of sarcopenia outside of the consensus definition established by the guidelines (muscle mass loss plus either weakness or impaired physical performance). There might also be important variations in the definition of obesity, with studies using high BMI, high body fat percentage, or high waist circumference. Thus, the definition of sarcopenic obesity is diverse and caution is warranted in the interpretation of studies on this topic.

The prevalence of sarcopenic obesity is predicted to be higher in patients with diabetes than in those without diabetes. Kim et al. reported that the percentage of skeletal mass to body weight was significantly lower and that the BMI and body fat percentage were the same or higher in patients with diabetes compared to those without diabetes [97].

Concurrent muscle loss and fat accumulation in sarcopenic obesity accelerate insulin resistance and inflammation [96], leading to atherosclerotic diseases. Insufficient insulin action suppresses myocyte differentiation, which can induce further muscle loss. This muscle loss leads to a reduction in basal metabolism, which induces further fat accumulation. Thus, fat accumulation and muscle loss can form a vicious cycle.

Various reports have shown an association between sarcopenic obesity and atherosclerosis. Atkins et al. reported that individuals with sarcopenic obesity, defined as high waist circumference and low midarm muscle circumference, were at a higher risk for cardiovascular mortality and all-cause mortality [98]. However, whether sarcopenic obesity is associated with higher mortality rate remains controversial.

Sarcopenic obesity is associated with functional disability. In a longitudinal study, Hirani et al. reported that the risk for frailty and ADL disability, which was significantly higher in individuals with sarcopenic obesity, was not significant in those with obesity alone [99]. Several studies reported that sarcopenic obesity was also associated with increased rates of falls [100] and incident fractures [101] in community-dwelling older persons, although the definition of sarcopenic obesity was different between the studies. However, these studies included individuals without diabetes and the impact of sarcopenic obesity on individuals with diabetes should be clarified.

No longitudinal studies to date have reported whether the incidence of cognitive impairment is higher in patients with sarcopenic obesity; however, some cross-sectional studies, including one comprising patients with diabetes [102], reported a significant correlation between sarcopenic obesity and impaired cognitive performance.

### 3.4. Malnutrition

Special attention should be paid to malnutrition in older adults with diabetes. Albeit not well established, several small studies have estimated that the prevalence of malnutrition or risk of malnutrition in elderly patients with diabetes is greater than 50% [103,104]. The rate of undernutrition, assessed by the Mini Nutritional Assessment (MNA), is higher in older adults with diabetes than in those without diabetes [105]. A survey of 1090 older hospitalized patients with diabetes in Spain, with a mean age of 78 years, found that 39.1% and 21.2% were at risk of malnutrition and had malnutrition, respectively [104]. Malnutrition, defined based on the MNA score, was associated with decreases in basic ADL, grip strength, physical performance of the lower limbs (stand-up test from the chair), and quality of life (QOL), longer hospital stays, and increased rates of institution and mortality [103,105,106].

Malnutrition can interfere with normal brain function and promote cognitive loss [107]. Additionally, the central role of malnutrition in the pathophysiology of frailty and sarcopenia is well supported [108,109], although glycemic control of patients with diabetes and malnutrition is often fair by chance due to decreased food intake. Around 40% of the proteins in human body is in skeletal muscles [110]. Muscle hypertrophy and atrophy depend on the balance between protein synthesis and degradation; therefore, the regulation of protein turnover is essential in muscle homeostasis. A diet rich in fruits and vegetables as well as exogenous antioxidant vitamins (such as vitamins E and C and carotenoids) and minerals can help restore the skeletal muscle redox homeostasis and prevent oxidative stress and the accumulation of reactive oxygen and nitrogen species, thereby contributing to muscle maintenance [111].

### 3.5. Changes in Body Weight

Unintentional body weight loss is another potential risk factor for cardiovascular diseases and mortality. In a secondary analysis of the ADVANCE study, an interventional trial of patients with diabetes, the risk for cardiovascular diseases and mortality doubled in those with unintentional body weight loss of > 10% compared to those without apparent body weight change (between −4% and 4%); the finding was independent of age [112]. Unintentional weight loss is a diagnostic criterion for frailty based on the CHS; therefore, body weight should be regularly monitored for early detection of even a slight weight reduction. In a cohort study of Korean patients with diabetes, not only low BMI but also significant changes in body weight (increase or decrease of more than 10%) were associated with increased risk for dementia [27]. Thus, abnormal increases in body weight should also be avoided in older adults with diabetes.

In contrast, intentional body weight loss achieved with lifestyle intervention can play a positive role in physical performance. In the Look AHEAD study, in which the effects of weight loss through lifestyle intervention, caloric restriction, and increased physical activity on overweight and obese middle-aged and older adults with type 2 diabetes were evaluated, those allocated to the lifestyle intervention group showed not only body weight reduction but also several other advantages, including higher physical function evaluated by the short physical performance battery and reduced risk for slow gait speed, compared with the control groups [113].

Evidence of associations of nutritional status and mortality, frailty, sarcopenia, and cognitive impairment is summarized in Appendix A. In summary, nutritional status which is not preferable for older patients with diabetes for the prevention of cognitive impairment, frailty and mortality is BMI ≥ 30 kg/m^2^, weight gain (>10%), MetS (<75 years of age), sarcopenic obesity, BMI < 18.5 kg/m^2^, unintentional weight loss, and malnutrition. These are shown in Figure 2.

## 4. Appropriate Energy and Macro/Micronutrient Intake in Older Adults with Diabetes

### 4.1. Energy Intake

Sufficient energy intake is recommended to reduce the incidence of frailty and mortality in older subjects. The European Society of Clinical Nutrition and Metabolism guidelines on clinical nutrition and hydration in geriatrics recommend an energy intake of approximately 30 kcal/kg body weight/day for older individuals [114], although additional considerations are indicated to individually adjust these values based on the nutritional status, physical activity level, disease status, and tolerance. It is reasonable that these recommendations should also be adopted by patients with diabetes. In the J-EDIT, Yoshimura et al. reported that lower energy intake was associated with lower BMI in older patients with diabetes and that the daily energy intake was significantly lower in those with a BMI of < 18.5 kg/m^2^ (average, 27.7 kcal/kg body weight/day) compared to those with a BMI of ≥ 18.5 kg/m^2^ (all groups average, ≥30 kcal/kg body weight/day) [115]. These results indicate that decreased energy intake in older patients with diabetes might lead to a lower BMI, which might be associated with an increased risk for frailty and mortality.

We assessed the six-year all-cause mortality rate in the J-EDIT participants by categorizing the cohort into quartiles according to energy intake/real body weight. The mortality risk was higher in Q1 (<24.85 kcal/kg body weight/day) and Q4 (≥34.79 kcal/kg body weight/day) than in Q2 (24.86–29.73 kcal/kg body weight/day) and Q3 (29.74–34.78 kcal/kg body weight/day) The risk of Q1 against Q3 was significant, whereas the risk of Q4 against Q3 was not. The lowest mortality risk was observed in Q2 comprising younger old patients (<75 years), whereas the lowest risk in older old patients (≥75 years) was observed in Q3 [116]. These results indicate that low energy intake is a mortality risk for older patients with diabetes and that energy requirement per actual body weight should be set higher in older old patients than in younger ones.

The 2019 diabetes care guidelines of the Japan Diabetes Society recommend that the total energy intake requirement in older adults should be calculated using age-dependent target body weight (kg) = (22–25 kg/m^2^ × height [m]^2^) multiplied by coefficients of physical activity [117]. Using a target body weight set based on age-dependent coefficients (23.5 and 25.0 kg/m^2^ for ages < 75 and ≥ 75 years, respectively), we found a U-shaped association between energy intake per target body weight and mortality similar to that reported for the association between energy intake per real body weight and mortality [116].

Several cross-sectional studies have investigated the association of energy intake with the incidence of frailty and cognitive impairment in older individuals. Lower total energy intake was associated with higher prevalence of sarcopenia [118] and cognitive impairment [119]. In the Rotterdam study, the risk of frailty decreased approximately by 5% with each 418.4 kJ (100 kcal) increase in total energy intake [120]. In that study, low energy intake but not low protein intake was associated with frailty.

However, in evaluating energy intake, caution is advised since food intake is frequently underreported by older individuals. In a Japanese cross-sectional study using calibration with doubly labeled water, the daily caloric intake by individuals at lowest risk for frailty was approximately 40 kcal/kg/ideal body weight (= 22 × height[m]^2^) [121]. Akin to the potential attenuation of the negative impact of high BMI in older individuals, as commented in the BMI section, sufficient energy intake is recommended for older patients with diabetes to prevent malnutrition except for morbidly obese individuals.

### 4.2. Proteins

Sufficient protein intake is also important to reduce the incidence of frailty or mortality in older individuals. The European Society of Clinical Nutrition and Metabolism guidelines recommend a minimum protein intake of 1.0 g/kg body weight/day; 1.2–1.5 g/kg body weight/day is recommended for older subjects with acute or chronic illness, although additional recommendations are provided for individual adjustments [108].

Observational studies have demonstrated that higher protein intake is associated with lower risk of incidence of frailty [122,123] and better lower-limb physical functioning and walking speed [124]. The results of interventional studies are controversial, which might be due to differences in participant background characteristics. Although a meta-analysis of randomized controlled trials (RCTs) of healthy older individuals revealed that protein or amino acid supplementation could not improve muscle mass or strength [125], another meta-analysis found that protein supplementation increased muscle mass in middle-old adults with and without sarcopenia [126]. Furthermore, in an interventional study including prefrail and frail patients, high protein intake improved muscle mass and physical performance [127]. Thus, higher protein intake may be recommended for patients who are at high risk for frailty and sarcopenia. Several studies have investigated protein supplementation in individuals with sarcopenic obesity. In an RCT of old women (aged > 65 years) with sarcopenic obesity, skeletal muscle mass was increased in those on a low-calorie/high protein diet, as compared with those on a low-calorie/low protein diet [128].

In a 3-year follow-up study of older adults with diabetes, the group receiving ≥ 1.0 g/kg body weight/day protein had less reduction in knee extension power and physical function compared to those receiving lower protein [129]. We recently reported that, in addition to low energy intake, low protein intake was a risk for mortality in older patients with diabetes. We analyzed the data pooled from the J-EDIT and the Japan Diabetes Complications Study (JDCS), which originally intended to validate the effect of lifestyle intervention on prevention of diabetic complications [130]. We found that those with low protein intake (<0.92 g/kg/body weight) showed significantly higher mortality, especially in those aged 75 years or older [131]. Since diabetes is a well-known risk factor for sarcopenia, sufficient protein intake should be recommended in older patients with diabetes except for those with end-stage renal failure.

Evidence related to cognitive function is lacking; however, cross-sectional [132] and prospective studies [133] indicate a positive association between protein intake and cognitive function. Therefore, future investigation is warranted to elucidate the potential association of energy intake with cognitive and physical function in patients with diabetes.

### 4.3. Vitamins

#### 4.3.1. Vitamin D

Vitamin D accelerates intestinal absorption of calcium, which increases bone mineral density and reduces fracture risk. Furthermore, vitamin D has a positive impact on muscular mass and strength [134]. Vitamin D deficiency correlates with cognitive decline [135], and vitamin D receptor is expressed in a number of brain regions including hippocampus [136]. In animal models, vitamin D has been shown to be important for neuronal maintenance and function [137]. A recent study reported that vitamin D might play a protective role in Alzheimer’s disease based on the enhanced clearance of amyloid beta in patients [138].

Several studies have reported that low serum vitamin D levels are associated with sarcopenia [139] and frailty [140,141]. Few studies investigating the impact of vitamin D supplementation alone on muscle mass and function [142,143] reported either no effect or beneficial results. Most of the RCTs evaluating vitamin D, administered with proteins or exercises, reported improvement in muscle mass or lower-extremity function in older patients with sarcopenia [144,145]. Although no study has targeted only patients with diabetes at high risk of sarcopenia, sufficient vitamin D intake together with sufficient protein intake and exercise might also be recommended in older adults with diabetes, especially in those with sarcopenia.

Regarding the role of vitamin D in cognitive function, a meta-analysis in 2017 reported a significant association between low serum or plasma vitamin D levels and worse cognitive function, with no beneficial effect observed following vitamin D supplementation [146]. A small RCT performed in patients with diabetes also showed that there was no positive effect on cognitive function in those supplemented with vitamin D [147].

#### 4.3.2. Other Vitamins

Vitamins C and E as well as carotene are known for their antioxidant activity. The B-vitamins folate, B_6_, and B_12_ are involved in the metabolism of homocysteine, which is involved in the pathogenesis of both atherosclerosis and Alzheimer’s disease.

Several observational studies have investigated the associations between the intake of these vitamins and frailty. For example, Balboa-Castilloa et al. reported that lower intake of vitamins B_6_, C, and E, and folate was associated with an increased risk for frailty [148]. Interventional trials with supplementation of these vitamins have hardly been conducted yet. Furthermore, there is no clear evidence showing the association between vitamin intake and frailty or functional disability in patients with diabetes.

Although B-vitamins and antioxidant vitamins can theoretically prevent cognitive impairment, observational studies on the association between vitamin intake and cognitive decline [149,150,151,152] and RCTs investigating the effects of vitamin supplementation [153,154] show inconsistent findings. The evidence is limited in patients with diabetes. In an observational study of the J-EDIT, we found that low intake levels of carotene, vitamin B_2_, pantothenate, calcium, and vegetables were associated with cognitive decline [28]. However, none of the RCTs [155,156,157] showed positive effects of the intervention with vitamin supplementation on cognitive function in patients with diabetes. However, vitamin supplementation in individuals with subclinical vitamin deficiency might have beneficial effects on cognitive function, which should be investigated in future studies including older patients with diabetes.

### 4.4. Fatty Acids

Polyunsaturated fatty acids (PUFAs), especially omega-3 fatty acids (ω-3FA) including eicosapentaenoic acid (EPA) and docosahexaenoic acid, can lower not only plasma triglyceride levels [157] but also the levels of inflammatory and oxidative markers [158].

A meta-analysis of RCTs showed no significant association between ω-3FA supplementation and incidence of cardiovascular events [159]. However, in a more recent RCT, supplementation with high-dose EPA (4 g/day) reduced the risk of cardiovascular events in patients already treated with statins for high triglyceride levels; this effect was also observed in the sub-analysis of older patients (≥65 years) with diabetes [160].

Fatty acid intake can also affect frailty and sarcopenia. In an observational study, the consumption of saturated fatty acids (SFA) was associated with a higher risk of frailty and mortality, whereas the consumption of PUFA and ω-3FA was associated with lower risk of mortality in patients aged over 50 years [161]. Two interventional studies on ω-3FA demonstrated their positive effects on muscle mass, grip strength [162], and walking speed in older individuals [163]; however, the sample size was small in both studies. In the only cross-sectional study of older (≥65 years) patients with diabetes, Okamura et al. reported that a reduced intake of ω-3FA was associated with a prevalence of sarcopenia defined by muscle mass and strength [164].

The meta-analyses of the many RCTs investigating the effect of ω-3FA on cognitive function reveal controversial findings. For example, although Zhang showed that a high intake of ω-3FA reduced cognitive decline [165], another study did not show significant effects of supplementation with ω-3FA, ω-6FA, or PUFA on cognitive function [153,166]. In an observational study of patients with diabetes, Devore showed that higher intake of SFA and trans-fatty acids and lower intake of PUFA were associated with cognitive impairment [167]. However, another RCT could not demonstrate the positive effect of ω-3FA supplementation on the prevention of cognitive impairment [168].

Based on these results, the intake of PUFA and ω-3FA may be important for the prevention of cardiovascular disease and sarcopenia rather than cognitive decline in older patients with diabetes.

Overall, although many observational studies suggest associations of various nutrients with frailty and cognitive impairment, interventional studies still fall short and most of the results are controversial. These diverse results might be accounted for by variations in participant background characteristics (e.g., age, race, geographical location, presence of frailty, sarcopenia, cognitive impairment, and nutritional deficiency). Special consideration is warranted to interpret these results, and further investigation in this field is necessary.

### 4.5. Mediterranean Diet and Healthy Diet

The Mediterranean diet, which is rich in vegetables, fish, olive oil, and wine, is known to reduce the risk of cardiovascular events and is widely considered as a healthy diet. In a meta-analysis of prospective studies, high adherence to the Mediterranean diet was shown to be associated with low all-cause mortality, even in older patients (≥65 years) [169].

Meta-analyses of observational studies reveal that high adherence to the Mediterranean diet is associated with a significantly reduced risk of frailty and functional disability [170,171]. Although data are limited in patients with diabetes, some reports indicated that adherence to the Mediterranean diet was associated with decreases in both the prevalence [172] and incidence [173] of frailty in older patients. Moreover, a similar association was observed particularly in those aged over 75 years in a cross-sectional study [172].

A meta-analysis of observational study revealed that high adherence to the Mediterranean diet was inversely associated with incidence of cognitive disorders [174]. Furthermore, although interventional studies are limited, Valls-Pedret et al. reported that cognitive functions were significantly improved in older subjects assigned to the Mediterranean diet plus olive oil or nuts [175]. An observational study of middle aged-old aged patients with diabetes reported that global cognitive function improved over the two-year observational period in those with high adherence to the Mediterranean diet [176].

In addition to the Mediterranean diet, recent studies have aimed to identify new categories in dietary patterns by analyzing the individual records of dietary intake. Among these, the healthy dietary pattern (i.e., healthy diet) is generally recognized as the high intake of fruit, vegetables, and whole grains and the low intake of meat, refined grains, sugar, and snacks (Figure 3) [177]. Unlike the Mediterranean diet, a healthy diet does not necessarily include olive oil and wine, making it a diet that applies to people all over the world. In a meta-analysis of longitudinal and cross-sectional studies including both the Mediterranean and non-Mediterranean countries, Fard et al. reported that high adherence to a healthy diet with higher intake of vegetables, fruits, and whole grains was associated with lower risk of frailty [177]. The World Health Organization guidelines for risk reduction of cognitive decline and dementia also recommend a Mediterranean-like diet and healthy, balanced food, both of which comprise sufficient quantity of fruits, vegetables, legumes, nuts, and whole grains with less sugars, fats, and salt, whereas they do not recommend supplements for each micronutrient [178]. The recommendation of a healthy diet over micronutrient supplementation is based on the cumulative beneficial effects of a healthy diet with lower risk of overdosing with micronutrient supplementation. Based on these results, although interventional studies fall short, the Mediterranean diet or a healthy diet should be recommended to reduce the risk of frailty and cognitive impairment in older patients with diabetes.

Evidence of associations of energy, macro/micronutrient intake, dietary pattern and mortality, frailty, sarcopenia, and cognitive impairment is summarized in Appendix A.

## 5. Strategy Shift in Dietary Management of Older Adults with Diabetes

### 5.1. Age-Related Differences in Nutritional Intake on Health Outcomes

The influence of nutritional intake on mortality and physical function varies with age. A survey of the general population found that the risk of all-cause and cancer mortality increased with increasing protein intake in those aged ≤ 65 years, whereas the risk of mortality increased with decreasing protein intake in those aged > 65 years [179]. In animal studies, the effect of high protein intake on mortality in young mice was due to the development and progression of tumors through the induction of growth factor receptor/IGF-1 signaling. In contrast, in very old mice, high protein diet reduced mortality whereas low protein diet remarkably reduced body weight and increased mortality [179]. The pooled analysis of the J-EDIT and the JDCS also demonstrated that the lowest quartile of protein intake was associated with highest mortality after adjustment for covariates in patients with diabetes aged 75 years or older but not in those aged younger than 65 years [131].

A meta-analysis reported that healthy dietary patterns (a diet high in fruit, vegetables, and whole grains) reduced the risk of frailty by 60% in those aged 65 years or older but not in younger people [177]. According to the J-EDIT, the dietary pattern was divided into three groups: healthy dietary pattern (abundant intake of vegetables and fish), snack dietary pattern (high intake of sweets, fruits, and potatoes), and oily dietary pattern (high intake of meat and fat). The mortality was lower in those with the healthy dietary pattern than in those with oily dietary pattern only among older patients aged over 75 years but not in those aged 65 to 74 years [180]. In addition, one study reported that patients with diabetes aged 75 years or older had improved walking and balance with improved adherence to the Mediterranean diet whereas a similar relationship was not observed in patients aged between 60 and 74 years [172]. Therefore, the dietary strategy of older people with diabetes should shift with advancing age from strict dietary restrictions for the treatment of MetS/obesity to diets for the prevention of frailty and sarcopenia. (Figure 4). In particular, a diet high in protein, vegetables, fruits, and fish is recommended in patients aged over 75 years.

### 5.2. Strategy Shift in Dietary Management from Metabolic Syndrome/Obesity to Frailty

There is no clear standard for the dietary strategy shift from MetS/obesity to frailty in older adults with diabetes. The strategy shift to frailty should be considered in those (i) aged 75 years or older, (ii) having frailty or sarcopenia, or (iii) having malnutrition (Figure 4). In such people, the strategy including (i) diet for the prevention of frailty, (ii) conducting resistance training or multi-component exercise, (iii) social participation and support, and (iv) optimal glycemic control should be implemented.

The diagnosis of malnutrition should be based on the Global Leadership Initiative on Malnutrition criteria (GLIM) [83]. The MNA and short-form MNA are useful screening tools for malnutrition [181,182]. In both cross-sectional and longitudinal studies, those at malnutrition risk according to the short-form MNA score were at higher risk of frailty, using both the Frailty Trait and the Frailty Index [183]. Low BMI, weight loss, and reduced food intake are important signs of malnutrition. It is necessary to be cognizant of not only the absolute low BMI value but also complaints such as weight loss and anorexia. The CHS criteria (three or more items) [8], clinical frailty scale [184], and the Kihon checklist (eight or more items) [185] can be used for the assessment of frailty. Sarcopenia can be diagnosed by evaluating muscle strength and muscle mass based on the Asia Working Group for Sarcopenia or the European Working Group on Sarcopenia in Older People 2 guidelines [186,187]. The Dementia Assessment Sheet for Community-based Integrated Care System-8 items (DASC-8) is an eight-item questionnaire based on IADL, BADL, and cognition [188], which has been used for the classification into the following three categories to determine glycemic targets in older Japanese adults with diabetes [5]: category I: intact cognitive function and no impairment of ADL, category II: mild cognitive impairment to mild dementia or impairment(s) of instrumental ADL, category III: moderate or severe dementia or impairment(s) of basic ADL or presence of multiple comorbidities or functional impairments. Because the prevalence rates of frailty and sarcopenia increase with more severe DASC-8 categories, those in DASC-8 categories II (11–16 points) and III (≥17 points) might be candidates for the prescription of a diet to prevent frailty [189].

For the prescription of a diet for older patients with diabetes, cognitive function, physical dysfunction (sarcopenia, frailty, IADL, and BADL), psychological state (depression), adherence to diet and medication, familial and social support, economic conditions, and comorbidities such as renal disease should be evaluated. If the patient does not have family support and cannot prepare their own meals, social resources such as home delivery services and day services at nursing homes should be secured. The assessment of renal function using estimated glomerular filtration rate is of great importance. Protein restriction for advanced chronic kidney disease (stage 4 or 5) in older patients with diabetes should be conducted with caution because it may lead to sarcopenia, frailty, and increased mortality. Among older patients aged over 65 years with chronic kidney disease, and not in younger patients, the risk of mortality was reported to be lower in patients with moderate protein intake (mean, 0.93 g/kg IBW/day) compared to those with low protein intake (mean, 0.70 g/kg IBW/day) [190]. It is necessary to frequently monitor renal function, skeletal muscle mass, muscle strength, and nutrition, and to determine whether prescribing a diet that prevents deterioration of renal function, frailty, or ADL disability is warranted.

Regarding nutritional therapy for frailty prevention, optimal energy intake and high protein intake (1.0–1.5 kg/kg actual weight) are recommended. High intake of vegetables (vitamins, A, C, E, and B-family) and fish (PUFA, ω-3FA, and vitamin D) may be desirable for the maintenance of physical and cognitive function. Healthy dietary patterns including the Mediterranean diet (Figure 3) are very important for increased survival as well as the prevention of frailty and dementia.

Dietary diversity (i.e., variety) is another approach that should be considered for frailty prevention. High dietary diversity was reported to be correlated with the intake of recommended nutrient-dense foods such as fruits, vegetables, and whole grains and non-recommended foods such as processed meats, salty snacks, and sweetened beverages [191,192]. High variety in diet as well as exercise was associated with a low incidence of frailty in a two-year longitudinal study [193]. Further studies on dietary variety and frailty in patients with diabetes are needed.

Diet-related QOL includes satisfaction with diet, burden of diet therapy, and perceived merits of diet therapy [194]. Dietary diversity is also related to diet-related QOL. In dietary management in older individuals, it is critical to maintain the diet-related QOL as well as good adherence.

As mentioned above, frailty and dementia have common mechanisms such as insulin resistance, hyperglycemia, hypoglycemia, arteriosclerotic disease, cerebral white matter lesions, malnutrition, and decreased physical activity (Figure 5). Therefore, effective prevention of frailty and dementia should include a common strategy including diet therapy, exercise including resistance exercise, optimal glycemic control, treatment of arteriosclerosis risk factors, social participation, and support.

Interaction between nutrition and hypoglycemic therapy should be noted in older patients with diabetes. Among oral antidiabetic drugs, users of sulfonylurea (SU) drugs and insulin are more likely to cause severe hypoglycemia in older adults, which leads to dementia, low ADL, incident cardiovascular disease, and increased mortality. Important precipitating factors of severe hypoglycemia in older adults are the decline in food intake along with an unchanged dose of SU drugs [195] or the administration of the wrong insulin product [196], and inappropriate diet or dietary timing, excessive alcohol intake, and sick days [197]. Low BMI [198] is also one of the risk factors for severe hypoglycemia. Attention should be paid to weight loss, which increases insulin sensitivity and vulnerability to hypoglycemia.

To avoid hypoglycemia, some treatment guidelines for older individuals with diabetes recommend less strict glycemic targets based on ADL, frailty, cognition, and dementia and set lower target limits such as a target HbA1c of < 7.0% [5,6]. In the standard care of older adults in the ADA, the simplification (deintensification) of complex regimens by reducing the dose or discontinuing medications in patients taking glucose-lowering medications is recommended if the individualized HbA1c target is maintained [199]. The structured educational program for patients and their families in case of a high risk for hypoglycemia, including the following items, is necessary to avoid hypoglycemia: (i) hypoglycemia unawareness or non-specific hypoglycemia symptoms, (ii) coping with excessive exercise, (iii) avoidance of skipping meals and heavy alcohol intake, (iv) conditions of high risk for hypoglycemia: low HbA1c value below the target limits, cognitive impairment, low ADL, weight loss, (v) coping with low food intake, vomiting or diarrhea: supplement with carbohydrate-containing food and drink, dose reduction or stopping of SU drugs, and adjustment of insulin dose.

Weight loss due to treatment with metformin, sodium-glucose cotransporter 2 inhibitors, and glucagon-like peptide-1 receptor analogs can lead to sarcopenia in older patients without adequate exercise. Vitamin B_12_ deficiency, which may lead to cognitive impairment, should also be noted in patients using metformin [200]. A low carbohydrate diet (40%) can increase the production of ketone bodies and euglycemic ketoacidosis in patients using sodium-glucose cotransporter 2 inhibitors [201,202].

## 6. Conclusions

In summary, insulin resistance, arteriosclerosis, brain white matter lesions, chronic inflammation, oxidative stress, and mitochondrial dysfunction are the common mechanisms of cognitive impairment and frailty in older adults with diabetes (Figure 5). Although the number of intervention studies is still scarce, diminished and excessive energy intake, low intake of protein, vitamin B-groups, vitamin D, and ω-3 FA are associated with frailty, cognitive decline, or increased mortality in epidemiological studies. A healthy dietary pattern consisting of optimal energy intake, high intake of protein, vegetables, and fish, including vitamin D and ω-3 FA, and low intake of meat, refined grains, sugar, and snacks could be an important dietary strategy to prevent frailty, cognitive decline, or mortality in older people with diabetes. Malnutrition, physical inactivity, social isolation, hyperglycemia, and hypoglycemia can be common risk factors for cognitive impairment and frailty. Therefore, the shift of dietary strategy to frailty prevention with advancing aging in combination with exercise, optimal glycemic and metabolic control, and social participation/support could extend healthy life expectancy and improve QOL in older adults with diabetes mellitus.

## Figures and Tables

**Figure 1 nutrients-12-03367-f001:**
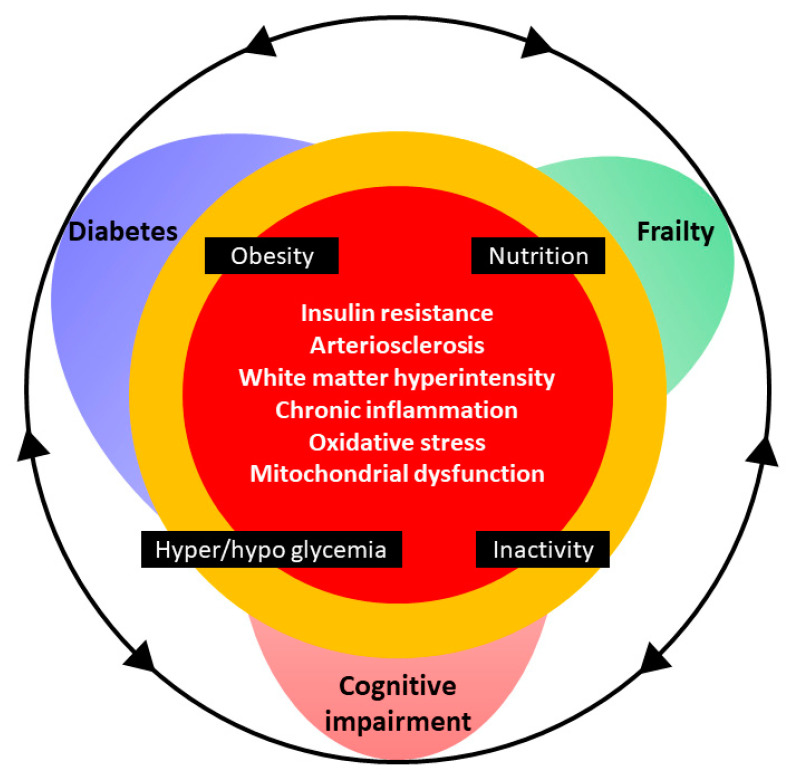
Cognitive impairment and frailty in older adults with diabetes. The mechanisms of diabetes, frailty, and cognitive dysfunction are closely related in the context of aging. Insulin resistance, arteriosclerosis, chronic inflammation, oxidative stress, and mitochondrial dysfunction are integrally involved in the pathogenesis of cognitive impairment, frailty, and diabetes mellitus. Obesity, inappropriate nutrition, physical inactivity, hyperglycemia, and hypoglycemia can be risk factors for cognitive impairment and frailty in older adults with diabetes. Proper management of body weight, nutrition, and exercise together with optimal glycemic control are the cornerstones of good health.

**Figure 2 nutrients-12-03367-f002:**
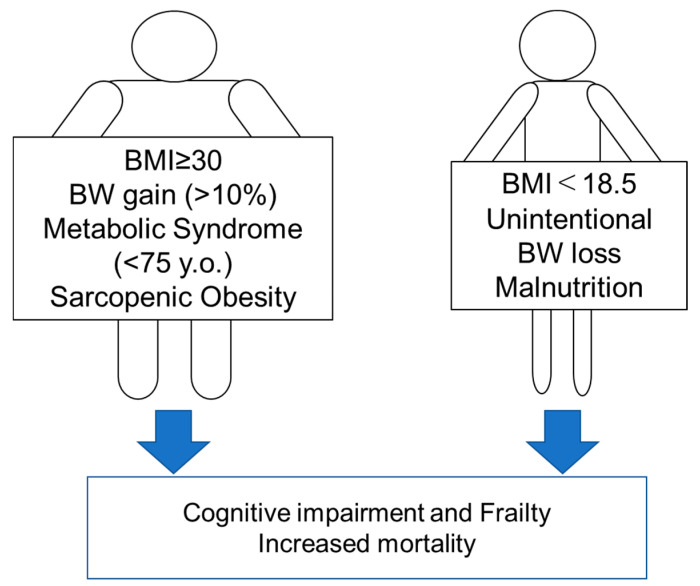
Nutritional states that are not preferred in older patients with diabetes for the prevention of cognitive impairment, frailty, and increased mortality. BMI, body mass index; BW, body weight.

**Figure 3 nutrients-12-03367-f003:**
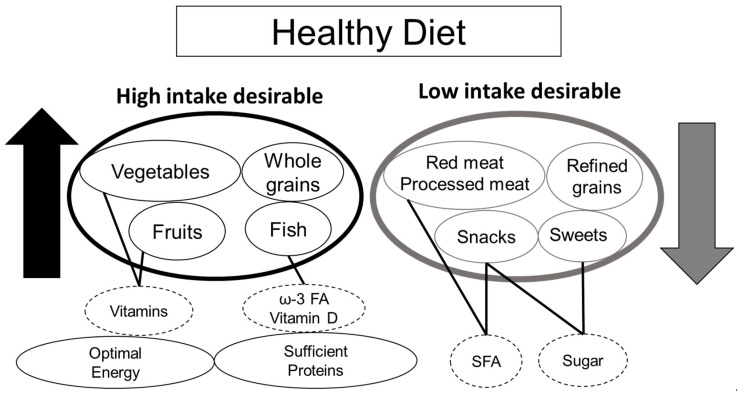
Healthy diets for older adults with diabetes. A healthy diet (Mediterranean-like diet) that is recommended for older patients with diabetes for the prevention of frailty and cognitive impairment. Food products that are recommended to be increased are indicated with black circles, whereas those recommended to be decreased are indicated in gray circles. Dashed circles indicate nutrients that are rich in each food product. ω-3 FA, omega-3 fatty acid; SFA, saturated fatty acid.

**Figure 4 nutrients-12-03367-f004:**
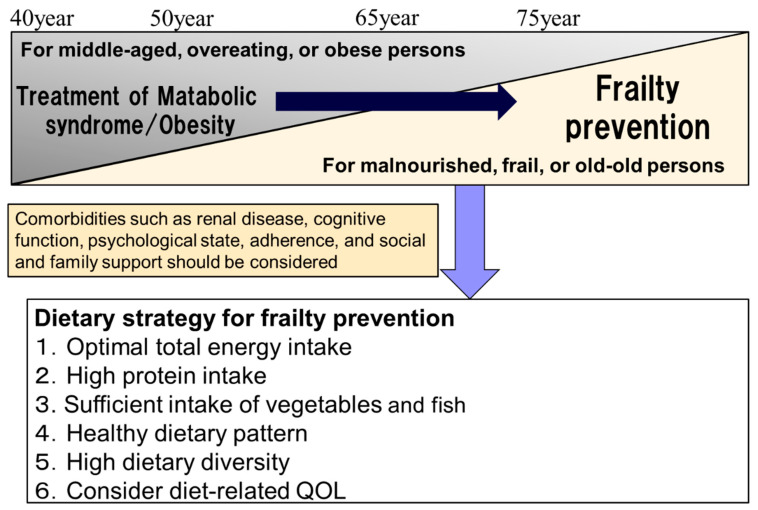
Shift in dietary strategies for frailty prevention in older adults with diabetes. For the prevention of frailty, optimal energy intake and high protein intake (1.0–1.5 kg/kg actual weight) are recommended. High intake of vegetables and fish may be desirable for the maintenance of physical and cognitive function. Healthy dietary patterns including a Mediterranean diet are highly recommended. High dietary diversity may be another approach for the prevention of frailty. In dietary management in older individuals, maintaining the diet-related quality of life is important. QOL, quality of life.

**Figure 5 nutrients-12-03367-f005:**
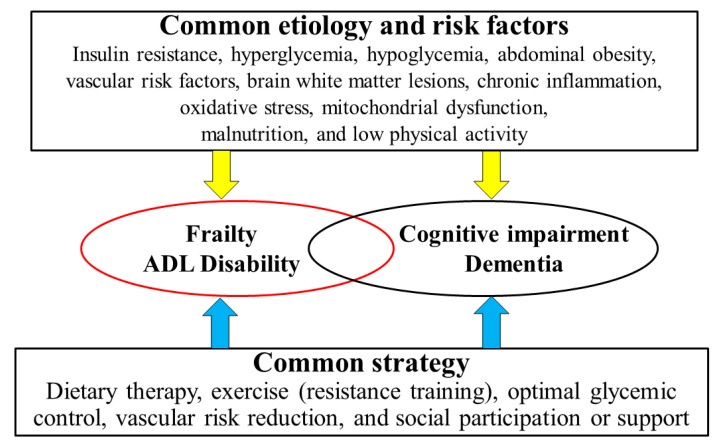
Common etiologies, risk factors, and strategies for frailty and cognitive impairment. Based on the common etiologies and risk factors shared between frailty and dementia, including insulin resistance, hyperglycemia, hypoglycemia, abdominal obesity, arteriosclerotic disease, cerebral white matter lesions, chronic inflammation, malnutrition, and low physical activity, common strategies such as dietary therapy, exercise, optimal glycemic control, treatment of arteriosclerotic risk factors, and social participation and support are important for effective management. ADL, activities of daily living.

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
