# Peer review of "Nutrition Management in Older Adults with Diabetes: A Review on the Importance of Shifting Prevention Strategies from Metabolic Syndrome to Frailty"

_nutrients, 2020, doi:10.3390/nu12113367_

Round 1
Reviewer 1 Report
This is a comprehensive review on non-pharmacological strategies on the prevention of frailty and cognitive decline associated with diabetes mellitus. The topic is of much interest for the readers of Nutrients, the article is in general well-written and clear, and the references are adequate. I suggest only minor changes before considering for publication.
- Even if the review is very thoughtful and ample, it is necessary to define an aim of the review. I suggest to add a brief introduction at the beginning outlining the points to be discussed and clearly stating the aim of the review.
- The authors cover many interesting topics but a summary at the end is essential to clarify the main findings and conclusions of the review.
- I suggest to include some tables summarizing the characteristics of the studies with strong evidence of some of the outcomes discussed.
Author Response
To #Reviewer1
Comments and Suggestions for Authors
This is a comprehensive review on non-pharmacological strategies on the prevention of frailty and cognitive decline associated with diabetes mellitus. The topic is of much interest for the readers of Nutrients, the article is in general well-written and clear, and the references are adequate. I suggest only minor changes before considering for publication.
- Even if the review is very thoughtful and ample, it is necessary to define an aim of the review. I suggest to add a brief introduction at the beginning outlining the points to be discussed and clearly stating the aim of the review.
Thank you very much for your valuable comments. According to the reviewer’s comment, we have written the introduction part in the beginning, which clarifies the aim of this review and standpoint for diabetic care in the elderly (Line 41-48).
- The authors cover many interesting topics but a summary at the end is essential to clarify the main findings and conclusions of the review.
Thank you for your comment. We have written the Conclusion section, again summarized the common etiology and risk factors for frailty and cognitive impairment in older diabetic patients, and described the common dietary strategy, especially in terms of nutritional intake and healthy diet pattern, for the prevention of frailty in older adults with diabetes mellitus (Line 737-750).
- I suggest to include some tables summarizing the characteristics of the studies with strong evidence of some of the outcomes discussed.
We have added two supplementary tables to summarize the evidence of previous studies. Supple Table1: Studies that investigated the associations between nutritional status and mortality, frailty, sarcopenia, and cognitive impairment; Supple Table 2: Studies that investigated the associations between energy, macro/micro-nutrient intake, dietary pattern and mortality, frailty, sarcopenia, and cognitive impairment.

Reviewer 2 Report
Overall, this is an interesting review suggesting a shift to preventing frailty in older adults with diabetes.
I would recommend the following revisions:
Line 191 – unclear sentence
Line 196 – you state that older adults are at greater risk of hypoglycemia. Please provide citations for this statement.
Line 242, I think you have a typo for the word ethic?
249-250 you state most of the studies and only go on to discuss one?
Line 373 – citations for this statement
Line 465 – what age group does old refer to?
Line 584 – unclear sentence
Fig 3 meat is misspelled
Line 598 & 599 sentence unclear
My most concerning content area is that in the last section you discuss the intersection between nutrition and hypoglycemic therapy but do not discuss that some classes of glucose-lowering medications are more likely to result in hypoglycemia as a side effect of the medication (e.g. sulphonylureas)
Additionally, the ADA has standards of care for older adults that bring up considerations such as avoiding hypoglycemia. I would suggest specifying how your recommendations differ from what has already been published on the treatment of older adults with diabetes, particularly from major organizations involved in publishing standards of care.
Author Response
To #Reviewer2
Comments and Suggestions for Authors
Overall, this is an interesting review suggesting a shift to preventing frailty in older adults with diabetes.
I would recommend the following revisions:
Line 191 – unclear sentence
Thank you for your valuable comments. We are confident that we can improve our manuscript. According to the reviewer’s comment, we have modified the sentence as follows:
The association between hyperglycemia and the effects of hyperglycemia on cognitive function were highly individualized (Line 201-202).
Line 196 – you state that older adults are at greater risk of hypoglycemia. Please provide citations for this statement.
According to the reviewer’s comment, we have added the following two references (Line 206, Ref):
[64] Abdelhafiz, A.H.; Rodríguez-Mañas, L.; Morley, J.E.; Sinclair, A.J. Hypoglycemia in older people - a less well recognized risk factor for frailty. Aging and disease 2015, 6, 156-167.
[65] Freeman, J. Management of hypoglycemia in older adults with type 2 diabetes. Postgraduate medicine 2019, 131, 241-250.
Line 242, I think you have a typo for the word ethic?
We apologize for this overlook. We have accordingly corrected the word as “ethnic (Line 252).”
249-250 you state most of the studies and only go on to discuss one?
According to the reviewer’s comment, we have cited the following two references here that indicate that there is no positive association between obesity (high BMI) and incidence of CHD or mortality, before mentioning the result of the J-EDIT study (Line 260-264, Ref).
[74] Zhang, X.; Shu, X.O.; Gao, Y.T.; Yang, G.; Matthews, C.E.; Li, Q.; Li, H.; Jin, F.; Zheng, W. Anthropometric predictors of coronary heart disease in Chinese women. Int J Obes Relat Metab Disord. 2004; 28, 734-740.
[75] Martínez-González, M.A.; García-Arellano, A.; Toledo, E.; Bes-Rastrollo, M.; Bulló, M.; Corella, D.; Fito, M.; Ros, E.; Lamuela-Raventós, R.M.; Rekondo, J.; Gómez-Gracia, E.; Fiol, M.; Santos-Lozano, J.M.; Serra-Majem, L.; Martínez, JA.; Eguaras, S.; Sáez-Tormo, G.; Pintó, X.; Estruch, R. Obesity indexes and total mortality among elderly subjects at high cardiovascular risk: the PREDIMED study. PLoS One. 2014, 9, e103246.
Line 373 – citations for this statement
According to the reviewer’s comment, we have added the following two references (Line 388, Ref):
[108] Verlaan, S.; Ligthart-Melis, G.C.; Wijers, S.L.J.; Cederholm, T.; Maier, A.B.; de van der Schueren, M.A.E. High Prevalence of Physical Frailty Among Community-Dwelling Malnourished Older Adults-A Systematic Review and Meta-Analysis. J Am Med Dir Assoc. 2017,18, 374-382.
[109] Sieber, C.C. Malnutrition and sarcopenia. Aging Clin Exp Res. 2019, 31, 793-798.
Line 465 – what age group does old refer to?
We have added the age group (aged >65 years) to the revised manuscript (Line 482).
Line 584 – unclear sentence
According to the reviewer’s comment, we have revised the sentence as follows:
The World Health Organization guidelines for risk reduction of cognitive decline and dementia also recommend a Mediterranean-like diet and healthy, balanced food, both of which comprise sufficient quantity of fruits, vegetables, legumes, nuts, and whole grains with less sugars, fats, and salt, whereas they do not recommend supplements for each micronutrient (Line 602-604).
Fig 3 meat is misspelled
We apologize for this overlook. The spelling has been corrected as “meat” (Fig 3).
Line 598 & 599 sentence unclear
According to the reviewer’s comment, we have revised the sentence as follows:
A survey of the general population found that the risk of all-cause and cancer mortality increased with increasing protein intake in those aged ≤65 years, whereas the risk of mortality increased with decreasing protein intake in those aged >65 years (Line 619-622).
My most concerning content area is that in the last section you discuss the intersection between nutrition and hypoglycemic therapy but do not discuss that some classes of glucose-lowering medications are more likely to result in hypoglycemia as a side effect of the medication (e.g. sulphonylureas)
Thank you for your valuable comment. According to your comment, we have added the description about the risk of hypoglycemia when administering SU drugs and insulin. We further mentioned the specific nutritional situation that can easily cause hypoglycemia when using these drugs and focused on these factors (Line 710-715).
Additionally, the ADA has standards of care for older adults that bring up considerations such as avoiding hypoglycemia. I would suggest specifying how your recommendations differ from what has already been published on the treatment of older adults with diabetes, particularly from major organizations involved in publishing standards of care.
Thank you for your valuable comment. We have cited the ADA recommendation of the simplification of glucose-lowering medications in older patients to avoid hyperglycemia. Then, we have further stated that it is highly important to educate patients and their family members in cases of high risk for hypoglycemia and described five critical points for educating them in the revised manuscript (Line 720-729).
Round 2
Reviewer 2 Report
Overall, the authors have presented a clear and comprehensive review.
The only comment that I still would recommend additional revision is outlined below:
Original comment for line 191 unclear sentence was in reference to this sentence directly below:
Hyperglycemia was associated with cognitive decline, and the increase of risk was different among 191 the people, impacting up to 50% [62].
The change now available in v2 appears to now be combined with what was originally a separate sentence and does not clear up any of the confusion
So these two sentences:
The association between hyperglycemia and cognitive function is well documented. Hyperglycemia was associated with cognitive decline, and the increase of risk was different among the people, impacting up to 50% [62].
Were combined into this:
The association between hyperglycemia and the effects of hyperglycemia on cognitive function were highly individualized [62].
This has actually now made the sentence even less clear as an association is a relationship between two things (you eliminated the other part of this sentence) and did not clear up the confusion for the sentence that was unclear.
I would recommend leaving the original first sentence (this one)
The association between hyperglycemia and cognitive function is well documented.
And try again to reword this second sentence so that it is clear
Hyperglycemia was associated with cognitive decline, and the increase of risk was different among the people, impacting up to 50% [62].
Author Response
Thank you very much for your valuable comment. We revised the sentence as below.
Hyperglycemia was associated with cognitive decline. The negative impact on cognitive performance was highly individualized, and approximately 50% of individuals with diabetes demonstrated such hyperglycemia-related cognitive disruption [62].(Line 201-203)